# Prolonged length of stays and its associated factors among pediatric surgical patients in Oromia, Central part of Ethiopia: A cross-sectional study

Redwan Nesha[1], Haymanot Mezmur[2], Yohannes Baye[3], Fentahun Meseret[3]*, Mulualem Keneni[3], Ayichew Alemu[3], Yalew Mossie[2], Henok Legesse[2]

1 School of Nursing, College of Health and Medical Sciences, Hawasa University Comprehensive Specialized Hospital, Hawassa., Ethiopia, 2 School of Nursing, College of Health and Medical Sciences, Haramaya University, Harar, Ethiopia, 3 Department of Pediatrics and Child Health Nursing, School of Nursing, College of Health and Medical Sciences, Haramaya University, Harar, Ethiopia

* mesie1055@gmail.com

## Abstract

### Background

Pediatric surgical diseases are conditions that require surgery in children to save lives, prevent disability, or provide palliative care. Surgeries can be major or minor based on factors like severity, and complexity. Prolonged hospital stay could significantly affect the limited resources of the hospital, and further lead to post-operative complications, and poor surgical outcome. Identifying the factors that contribute to longer hospital stays could improve quality of patient care and service delivery. However, there is limited evidence in Ethiopia, particularly among paediatrics population. Hence, this study aimed to assess the length of hospital stay and its associated factors among pediatric surgical patients admitted to Adama Comprehensive Specialized Hospital and Asella Teaching Hospital, Oromia, Ethiopia.

### Methods

Institutional based cross-sectional study was conducted through secondary data extraction from February 1, 2023, to January 30, 2024. A simple random sampling technique was employed to select 422 pediatric patients admitted to the surgical units. SPSS version 26 was utilized for analysis and data was presented in frequency, percentage, tables and figures. The association between independent variables and dependent variable (Prolonged Length of Hospital Stay) was assessed using binary logistic regression. A length of hospital stays for more than or equal to $75^{th}$ percentile (≥ 12.5 days) was considered as prolonged. Finally, variables with p-value <0.05 were considered statistically significant.

**Data availability statement:** All relevant data are within the paper.

**Funding:** The author(s) received no specific funding for this work.

**Competing interests:** The authors have declared that no competing interests exist.

**Acronyms:** ACSH: Adama Comprehensive Specialized Hospital; ARTH: Asella Referral and Teaching Hospital; ASA: American Society of Anesthesiologists; BUPA: British United Provident Association; CEPOD: The Confidential Enquiry into Patient Outcome and Death; EPCO: The European Perioperative Clinical Outcome; HAI: Hospital Acquired Infection; LOS: Length of Stay; PLOS: Prolonged Length of Stay; WHO: World Health Organization.

## Results

The prevalence of prolonged length of stay was found to be 19.6% [95% CI: 15.8%, 23.4%]. Undernutrition [AOR: 2.6; 95% CI: 1.11,6.08], children admitted due to trauma [AOR: 4.1; 95% CI: 1.75, 9.83], and postoperative complications [AOR: 7.8; 95% CI: 3.99, 15.6] were positively associated with the outcome variable. Whereas, emergency admission [AOR: 0.30; 95% CI: 0.14, 0.61], was negatively associated with prolonged length of stay.

## Conclusion

Approximately one-fifth of the participants experienced prolonged in hospital stays. Therefore, clinical practitionars, hospital administrators and respective stackholders could significantly reduce the childrens length of stay and utilize hospital resources effectively through screening nutritional status, keen attention to trauma admission and monitoring and managing post-operative complications as early as possible. Moreover,ensuring all children to have food security may result in shorter stay in the hospital. Expansion of surgical work force could also improve patient outcome following surgery among paediatrics population.

## Introduction

Pediatric surgical disease encompasses various surgical pathologies and disorders in the infancy to adolescent age group, which equires surgical intervention [1]. The incidence of surgical disorders, oscillates based on age, time, anatomical location in which the problem originated and other sociodemographic factors [2,3];where surgically treatable childhood diseases roughly accounts for up to 30% in a global context [3]. In pediatric surgery units, the most common causes are acute abdomen, followed by urological emergencies, trauma, congenital defects, foreign bodies, and infectious disorders like croup, intestinal parasites, and abscesses [4–6].

Surgeries are categorized into major and minor based on the severity of the condition, affected body parts, surgerical complexity, and recovery time. Major surgeries, like heart surgery, head, neck, and chest may require a hospital stay of several weeks;whereas, minor surgeries like appendectomy and hernia repairs may require few days to stay in the hospital. However, the length of hospital stay may vary based on the child's age, health status/presence of comorbidity, type of surgery and other varies factors including quality of service delivery [7].

The length of hospital stay (LOS) refers to the duration a patient spends in the hospital after admission [8];moreover,e prolonged length of hospital stays (PLOS) can be declared when the total number of bed days a patient spends in the hospital for surgical care that exceeds their expected duration [9]. Longer stays lead to higher resource consumption, hospital-acquired infections (HAIs), complications, and mortality rates;it also delays timely treatment for critically ill patients, causing a shortage of capacity [10].

From the global disease of burden around 28–30% of cases were surgically repairable, where only 3.5% of 234 million surgeries have been performed in developing nations including Africa [6,11]. In 2017, World Health Organization (WHO) reported that, 1.7 billion children were unable to access life-saving surgical care, leading to 0.02% pediatric mortality, 13.9% complications, and 5.7% serious adverse events, even following the intervention due to unmet needs and deliance [1,12]. Sub-Saharan Africa faces the world's largest unmet surgical care need, accounting for 29% of global unmet needs with an estimated 85.4% cumulative risk of further complications in all surgical diseases under the age of 15 years [13]. Similarly, Ethiopia faces a significant unmet need for pediatric surgical problems, accounting for 31% of surgical management and 33% of pediatric hospital admissions [14]. The current practice of pediatric surgery need to be evaluated with special attention, due to an increase in hospital stays record across the country sofare [15]. The duration of patients' hospital stays is a crucial metric for evaluating quality patient care in hospitals with respect to the nature of the problem [11]. For example,children with congenital anomalies had higher hospitalization rates (25–31%) compared to those without them [16].

Pediatric surgical patients in low- and middle-income countries face longer hospital stays due to malnutrition, surgical site infection, and postoperative stress, which can again lead to higher post-operative complications and death [17]. In the less developed nations, the most common reasons for hospitalization has been reported to be infections, osteomyelitis, traumas, and burns, with a range of 6.5–46 days length of hospital stay; which is relatively higher compaired with developed nation (3.4 days) [18]. Children's duration of hospitalization can have an impact on the proper utilization of healthcare resources in the hospital [19]. Furthermore, prolonged hospital stays can be emotionally, economically, and psychologically challenging for both the child and their family [20,21].

Identifying factors affecting the length of stay is crucial for enhancing treatment, better surgical outcomes, effective hospital resource utilization, enhanced patient safety& satisfaction, and lowering possible financial burden [22]. Factors affecting pediatric surgical hospital stays could be broadly categorized as socio-demographic &patient-related factors and clinical-related such as pre-operative, intra-operative & post-operative factors [9,11,13,15,23]. As a result, perioperative care with scientific evidences have a paramount benefit for children, clinical practitioners and health care administrators in the monitoring and management of pediatrics surgical cases to shorten the length of stay in hospital [7,24]. However, a paucity of evidence was existed in Ethiopia. Existing studies were conducted on medical cases, emphasized more on adult surgical population and factors associated with PLOS were less explored;identifying these factors will help to develop evidence based practice aimed at reducing the length of hospital stay among pediatrics population admitted due to surgery.

Therefore, the objective of this study was to assess length of hospital stay and its associated factors among pediatric surgical patients admitted at teaching Hospitals in oromia central part of Ethiopia.

## Methods and materials

### Study setting and period

The study was conducted in Oromia, central Ethiopia in two selected teaching hospitals: Adama Comprehensive Specialized Hospital (ACSH) and Asella Referral and Teaching Hospital (ARTH). ACSH, was established in 1938, is one of the leading medical hospital in central Ethiopia with 75 pediatric beds, 15 of which are dedicated to give services for the patient with surgical and orthopedic problems. The average number of pediatric surgeries performed in a year in ACSH is 879, with an average of 73 patients per month. The second hospital, ARTH, is located in the Arsi region, serving residents of the eastern and southern regions of Ethiopia. It was established in 1964, it offers outpatient, inpatient, emergency, ambulatory, and other specialized services. The study encompassed a secondary data review from February 1, 2023, to January 30, 2024. The data extraction period was from April 1st to 30th, 2024

### Study design

A hospital-based cross-sectional study was conducted.

## Source population

All pediatric surgical patients who were admitted to Adama Comprehensive Specialized Hospital and Asella Referral and Teaching Hospital.

## Study population

All pediatric surgical patients admitted at Adama Comprehensive Specialized Hospital and Asella Referral and Teaching Hospital during the study period.

## Inclusion and exclusion criteria

All pediatric surgery patients admitted to Adama Comprehensive Specialized Hospital, as well as to Asella Referral and Teaching Hospital were included. However, Neonates and medical records missing relevant data (clinical,category of surgery/minor Vs major and discharge outcomes including length of stay) were excluded from this study.we have excluded neonates because of neonatal and pediatric surgery differ significantly the unique physiological characteristics and vulnerabilities of newborns(neonates) compared to older infants and children

## Sample size determination

Due to the lack of previous studies done exactly analogous to our title,we have decided to use 50% as the population proportion prevalence value of Prolonged Length of Stay in pediatric surgery patients with the standard normal distribution ($Z\alpha/2 = 1.96$), a confidence interval of 95%, and $\alpha = 0.05$, $P = 50.0\%$; d or a tolerable margin of error $= 0.05$ and the sample size was calculated by using a single population proportion formula as of the following

Where $p =$ the proportion of ProlongedLength of Stay; $q =$ proportion of less than the average length of stay; $d =$ margin of error n$= (1.96)2*.5*.5/ (0.05)2$ This gives 384, by adding a 10% non-response rate, the final sample size was 422.

## Sampling procedure and technique

Two public hospitals, Adama Comprehensive Specialized Hospital and Asella Teaching and Referral Hospital were selected purposely; to allocate the number of study participants from pediatric surgery at ACSH and ARTH, a one-year pediatric surgical ward patient record was assessed and the trend showed that, 1275 (879 from ACSH and 396 from ARTH) were admitted in the year of 2022. Based on probability proportional to size sampling technique, the sample size distribution was determined to be 291 patients from ACSH and 131 patientsfrom ARTH. (proportionally allocated to both hospitals based on their annual surgery report) Then, the participants were selected by computer-generated simple random technique using their Medical Registration Number. (**Fig 1**).

## Data collection techniques and procedures

Data were collected using a pretested and structured questionnaire that has been developed through critically reviewed literatures and internationally accepted scales. The Confidential Enquiry into Patient Outcome and Death (CEPOD), British United Provident Association (BUPA) and American Society of Anesthesiologists (ASA) classification were used to determine the type of admission, the magnitude of the surgery and the physical status respectively. [11,14,23,25–31]. The questionnaire consists of five sections: socio-demograhic characteristic (8 items); pre-operative factors (16 items); intra-operative factors (11 items); post-operative factors (8 items); and surgical outcome (6 items)

Data were collected by six BSc nurses (three for each hospital) with one supervisor having MSc in Pediatrics and Child Health Nursing. First, the target study populations were identified from pediatrics surgical registration books in the wards. Next, using the Medical Registration Number (MRN) from the registration book the patient's medical records were retrieved. Then, data extraction was made using predefined questionnaire/checklist sublimented by maternal interview. Incomplete medical records were replaced with additional random sample..

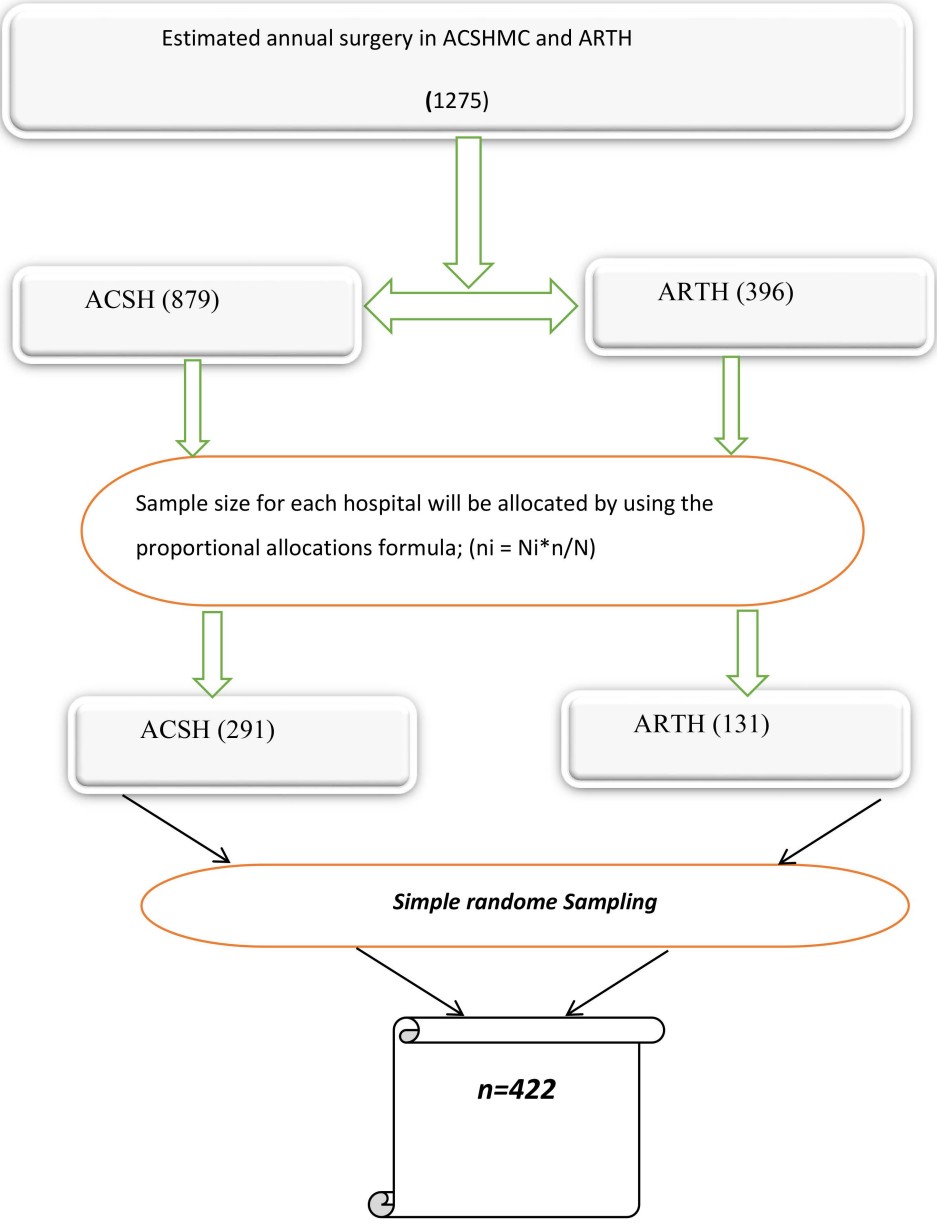

**Fig 1. Schematic presentation of sampling technique to assess Prolonged length of stays and its associated factors among pediatric surgical patients in oromia, central Ethiopia, 2024.**

## Dependent and independent variables

The prolonged length of hospital stay (PLOS) is the dependent variable. Whereas, socio-demographic-related factors (age, sex, residency, religion, and medical insurance); pre-operative factors (comorbidity, nutritional status, prophylaxis medication, medication, andanemia); and intraoperative and postoperative factors (type of operation,anesthesia postoperative complication, blood transfusion, and complexity of the surgical procedure) were the independent variables.

## Operational definitions

**Prolonged Length of Hospital Stay (PLOS)** The total number of days a patient spends in bed during a hospital stay that exceeds the expected length of stay for a given procedure. **PLOS** is defined as the proportion of the study participants with hospital stays greater than or equal to the 75th percentile ofthe entire study population [9,32,33]. [the 75th percentile cutoff point was suggested and used in this study, for minimizing procedure specifc LOS variations and possible outliers[933 34]. In this study, it was considered to be ≥ 12.5 days.

The **physical status level:** the updated American Society of Anesthesiologists (ASA) physical status classification was used. In which it's classified as: ASA I: Health children with no acute or chronic disease; ASA II: child with mild systemic disease; ASA III: child with severe systemic disease with substantial functional limitation; ASA IV: child with severe systemic disease that is constant threat of life; ASA V: brain dead patient [34]; **Malnutrition/undernourished:** it is based on physician diagnosis with 'Yes' or 'NO' response: Wh/ht/length, MUAC and BMI for age were found to be the most utilized indices to indicate the presence of malnutrition with the value below standard deviation (SD) based on WHO recommendation as cut of point [35]; **Co-morbidity:** Implies concomitant diseases that are not the complications of the surgery; **Prophlaxis medications:** are medications that are concurrent drugs prescribed for the treatment or prevention of infection before surgery as a preoperative care; **Elective surgery**: the term for operations planned in advance; **Emergency surgery**: the term used for operation that requires immidate admission to hospital to be performed within 24 hours; **Complexity of the surgical procedure:** To evaluate surgical complexity, the surgeon examined the association between 3 factors that are surrogates of surgical level-estimated total operative time, bleeding, and complications [36]; **Anemia** (in g/dl) was defined according to the WHO criteria as of the following: children 6-59 months of age (Mild, 10.0-10.9; Moderate, 7.0-9.9; Severe, <7.0); children 5-11 years of age (Mild, 11.0-11.4; Moderate, 8.0-10.9; Severe, <8.0); children 12-14 years of age (Mild, 11.0-11.9; Moderate, 8.0-10.9; Severe, <8.0) [37,38]; **Low white blood cell count**: A white blood cell count below 1000 cells per microliter [39]; **Pediatrics/children**:consider the age group of less than 18 years; **Intra-operative adverse events:** Intra-operative adverse events are defined as a deviation from the ideal intra-operative course from skin incision to skin closure including surgery and anesthesia-related events. Classintra classification grading system was used which ranges from grade 0, no adverse events, to grade V, intra-operative death [40]; **Post-operative complications:** The European Perioperative Clinical Outcome (EPCO) definitions were utilized. The definitions provide 22 individual clinical adverse events including healthcare-associated infections (HAIs), surgical complications, and respiratory complications [41,42].

## Data processing and analysis

The data were collected using KoboToolbox and then transferred into SPSS version 26 software for analysis. The results were organized, summarized, and presented using appropriate descriptive measures. The outcome variable was categorized using 75th percentile as a cut-off value. Those participants who stayed more than or equal to 75th percentile were categorized as PLOS and those who scored below were catagorized as expected length of stay. A binary logistic regression model was employed to identify factors associated with PLOS. Variables with a p-value of less than 0.25 in the bi-variable analysis were included in multi-variableanalysis.The model's goodness of fit was assessed using the Hosmer and Lemeshow test (p = 0.8348). Multicollinearity was evaluated through the Variance Inflation Factor (VIF), with each independent variable showing a VIF of less than 10 and a mean VIF of 1.38, indicating no evidence of multicollinearity. Finally, Statistical significance was declaredat a p-value of less than 0.05 with an Adjusted Odds Ratio (AOR) and 95% Cofidence Interval (CI).

## Data quality control

Data collectors received two days of training on the data collection's aims, contents, and processes. The questionnaire was written in English first, then translated by language experts into Afan Oromo and Amharic, and subsequently back into English to ensure consistency. Prior to data collection, the questionnaire was pretested on 5% of the sample (n = 21) in the

same study area for clarity, completeness and availability of measured variables;accordingly necessary modification was made. The Supervisor and investigators reviewed the data for completeness throughout the data collection process.

### Ethical approval and consent to participate

Before starting the data collection process, the study protocol was approved by Haramaya University, College of Health and Medical Sciences, Institution Health Research Ethical Review Committee (IHRERC)/Ref No. IHRERC/038/2024; An official letter of cooperation was submitted to Adama Comprehensive Specialized Hospital and Asella Referral and Teaching Hospital's head to obtain cooperation and consent to facilitate the study. Voluntary, informed, written, and signed consent was waived from the head of the hospitals on behalf of all study participants/subjects and/or their legal guardians. This study had no danger or negative consequences for the study participants. Confidentiality of information was assured by excluding names and identifiers in the questionnaire. Access to collected information was limited to the principal investigator and confidentiality was preserved throughout the time This study was conducted in accordance with the guidelines and regulations of the Declaration of Helsinki.

## Results

### Sociodemographic characteristics

A total of 422 participants were included in the study. The mean (SD) age of the participants was 6.40 (±4) years, and about 183(43.4%) were aged between 5 and 10 years. The majority 272 (64.4%) of the participants were males, while the rest 150 (35.6%) were females. Regarding religion 205 (48.6%) were from the Muslim religion's follower's family, and 220 (52.1%) were urban dwellers. Many of them 283 (67.9%) had no health insurance.(**Table 1**).

### Pre-operative characteristics

About half, 208 (49.3%), of the participants, were admitted for emergency surgery and the majority of the participants, 296 (70.8%) had ASA I physical status level at presentation. Gastrointestinal disorders were the main reason of admission for about 171 (40.5%) patients, followed by congenital anomalies (31.3%). Among the gastrointestinal (GI) disorders, appendicitis is the leading cause of admission (57.9%), followed by intussusceptions (14.6%) and intestinal obstruction (5.8%). Only twelve participants (2.8%) had comorbidity. Nearly eleven percent of the participants (n = 45) were malnourished, and

**Table 1. Sociodemographic characteristics of the study participants Admitted To Two Teaching Hospitals In Central Ethiopia,2024(n = 422).**

| Variable | Category | Frequency | Percent |
|---|---|---|---|
| Age | <5 | 162 | 38.4 |
| | 5-10 | 183 | 43.4 |
| | >10 | 77 | 18.2 |
| Sex | Male | 272 | 64.4 |
| | Female | 150 | 35.6 |
| Religion of parents/guardian | Muslim | 205 | 48.6 |
| | Orthodox | 119 | 28.2 |
| | Protestant | 94 | 22.3 |
| | Others | 4 | 0.95 |
| Residence of parents/guardian | Rural | 202 | 47.9 |
| | Urban | 220 | 52.1 |
| Medical insurance | Yes | 134 | 32.1 |
| | No | 283 | 67.9 |

24 (5.7%) had anemia during the preoperative period. The mean waiting time to seek medical attention in the hospital was 1.3 hours and 50 (11.9%) of the participants had a white blood cell (WBC) count of less than 5000 μ/l.(**Table 2**).

## Intra and post-operative characteristics

The majority of the surgical procedures, 364 (86.3%) were major surgeries, and 94.08% of the procedures were conducted under general anaesthesia. Among the major procedural specialties orthopedics (21.6%) and laparotomies (14.4%) were found to be the predominant. About 294 (69.7%) of the surgeries have used the WHO safe surgical checklist. The mean duration of anaesthesia was 43.3 (±59.6) minutes, while the average duration of stay in surgery was 36.9

**Table 2. Preoperative related characteristics of pediatric surgical patients Admitted To Two Teaching Hospitals In Central Ethiopia,2024 (n = 422).**

| Variable | Category | Frequency | Percent |
|---|---|---|---|
| Type of admission | Elective | 144 | 34.1 |
| | Emergency | 208 | 49.3 |
| | Scheduled | 51 | 12.1 |
| | Urgent | 19 | 4.50 |
| ASA physical status classification | I | 296 | 70.8 |
| | II | 115 | 27.5 |
| | III | 3 | 0.72 |
| | IV | 4 | 0.96 |
| Admission diagnosis | Gastrointestinal | 171 | 40.5 |
| | Congenital surgical problem | 132 | 31.3 |
| | Injury | 59 | 14.0 |
| | Genitourinary surgery | 14 | 3.3 |
| | Burn | 5 | 1.2 |
| | Endocrine disorders | 2 | 0.5 |
| | Others | 39 | 9.2 |
| Gastrointestinal disorders | Appendicitis | 99 | 57.9 |
| | Intestinal obstruction | 10 | 5.8 |
| | Intussusceptions | 25 | 14.6 |
| | Pyloric stenosis | 2 | 1.1 |
| | Rectal prolapsed | 4 | 2.3 |
| | Other | 31 | 18.1 |
| Comorbidity | Yes | 12 | 2.8 |
| | No | 410 | 97.2 |
| Hydrocephalus | Yes | 3 | 0.7 |
| | NO | 419 | 99.3 |
| Preoperative anemia | No | 396 | 94.3 |
| | Yes | 24 | 5.7 |
| Nutritional status at admission | Normal | 377 | 89.3 |
| | Undernurshied | 45 | 10.7 |
| Waiting time to seek medical attention in hospital | 1.31(±2.33) hrs | | |
| Preoperative Platelet count | $\geq 150 \times 10^3$ | 399 | 94.55 |
| | $< 150 \times 10^3$ | 23 | 5.45 |
| Preoperative WBC count | $\geq 10^5$ | 372 | 88.15 |
| | $< 10^5$ | 50 | 11.85 |

(±48.2) minutes. One in ten (13.5%) of participants have developed postoperative complications; of these, 61.4% were found to be healthcare-associated infections (HAIs)and 26.3% have developed sepsis. Among patients who developed HAIs, 82.9% and 17.4% account for surgical site infection and hospital-acquired pneumonia respectively.(**Table 3**).

**The prevalence of Prolonged Lenght of Hospital stay (PLOS)**

In this study, the 75th percentile length of hospital stay was 12.5 days, and the prevalence of prolonged LOS (≥ 12.5 days) was found to be 19.6% [95% CI: 15.8%, 23.4%]. The median length of stay in the hospitals was 6 days (IQR = 7), and the mean LOS was 8.6 days.(**Fig 2**).

**Table 3.** Intra And post-operative related characteristics of pediatric surgical patients Admitted To Two Teaching Hospitals In Central Ethiopia,2024(n = 422).

| Variable | Category | Frequency | Percent |
|---|---|---|---|
| WHO safe surgical checklist | Present | 294 | 69.7 |
| | Absent | 128 | 30.3 |
| BUPA scoring | Minor | 26 | 6.2 |
| | Major | 364 | 86.3 |
| | Major plus | 32 | 7.5 |
| Type of surgery | Head and neck(bar hole,craniotomy) | 26 | 6.16 |
| | Anoplasty | 17 | 4.03 |
| | Laparotomy | 61 | 14.4 |
| | Orthopedic | 91 | 21.56 |
| | Thoracic | 23 | 5.45 |
| | Urologic | 2 | 0.48 |
| | Others | 202 | 47.87 |
| Prophylaxis | Yes | 350 | 82.9 |
| | No | 72 | 17.1 |
| Type of anesthesia | General anesthesia | 397 | 94.08 |
| | Local anesthesia | 8 | 1.90 |
| | sedative anesthesia | 17 | 4.03 |
| Duration of anesthesia | 43.3 (±59.6) minutes | | |
| Duration of surgery | 36.9 (±48.2) minutes | | |
| Postop compliaction | Yes | 57 | 13.5 |
| | No | 365 | 86.5 |
| Type of complication | HAIs | 35 | 61.40 |
| | Sepsis | 15 | 26.32 |
| | Shock | 1 | 1.75 |
| | Wound dehiscence | 6 | 10.53 |
| HAIs (n = 35) | Pneumonia | 6 | 17.14 |
| | SSI | 29 | 82.86 |
| Blood transfusion | Yes | 64 | 15.2 |
| | No | 358 | 84.8 |
| Woundcare per day | Once | 332 | 78.67 |
| | Twice | 72 | 17.06 |
| | Three times | 15 | 3.55 |
| | Every other day | 3 | 0.71 |

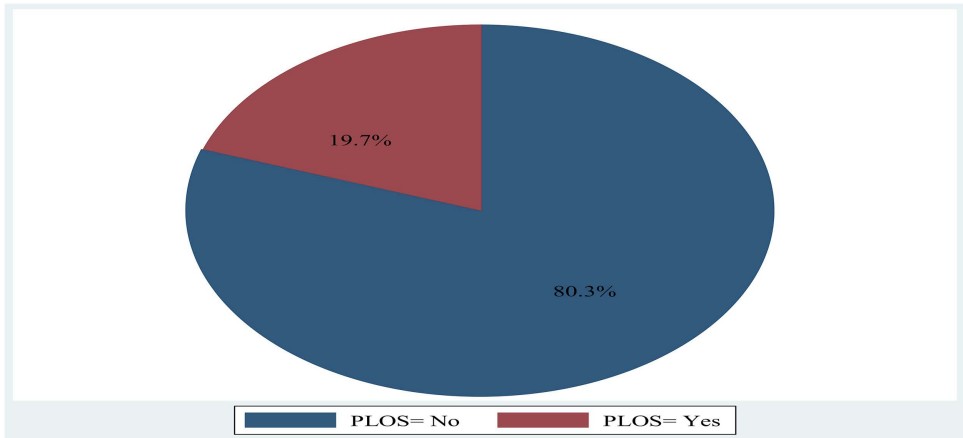

**Fig 2. The prevalence of PLOS among pediatric surgical patients admitted to two teaching hospitals in Oromia,Central part of Ethiopia, 2024 (n = 422).**

## Factors associated with PLOS

In bi-variable analysis, 10 variables were found to be candidates for multi-variable logistic regression; these variables were age, residency, comorbidities, type of admission, preoperative anemia, undernutrion, admission diagnoses, health insurance, postoperative complications, and utilization of the WHO safe surgery checklist (p-value <0.25). However,in multi-variable logistic regression, 4variables namely; types of admission, undernutrition, admission diagnosis and postoperative complications were significantly associated with PLOS at a p-value of <0.05.

Emergency admissions were 70% [AOR: 0.30; 95%CI: 0.14, 0.61] reducing the pronged LOS in the hospital as compared to elective admissions.Undernourished children who underwent surgery were 2.6 [AOR:2.6; 95% CI: 1.11, 6.08] times more likely to stay longer in the hospital as compared with those children with normal nutritional status.

The odds of prolonged LOS for children admitted due to trauma were 4.1 [AOR: 4.1; 95%CI: 1.75, 9.83] times higher than those with gastrointestinal disorders. Additionally, children who developed post-operative complications had approximately 8 [AOR: 7.8; 95%CI: 3.99,15.6] times higher odds of prolonged LOS compared to those who did not develop complications(**Table 4**).

## Discussion

The study aimed to assess the prevalence of prolonged length of hospital stay and its associated factors among pediatric surgery patients admitted to Adama Comprehensive Specialized Hospital (ACSH) and Asella Referral and Teaching Hospital (ARTH), Oromia, Central Ethiopia. The prevalence of PLOS in this study was 19.6% [95%CI: 15.8%, 23.4%]. Emergency admission, undernutrition, admission diagnoses, and postoperative complications were significantly associated with PLOS.

Despite existing literature specifically focused on PLOS among adult populations and some specific pediatric surgerical procedures and a paucity of evidence existed for comparison, it is valuable to provide an overview of the findings concerning studies conducted in other contexts on the length of hospital stays.

The Prevalence of PLOS in our study was in line with a study conducted in Canada (22.6%) [22] and in the University of Cincinnati College of Medicine (16.4%) [43]. This might be due to the complexity and recovery time of the surgical procedures lead to extended and comparable LOS with this study finding. However, this study's finding was found to be higher than a study conducted in USA (12.2%) [44]. This variation might be due to the specified procedure recovery time

**Table 4. Factors associated with PLOS among pediatric surgical patients Admitted To Two Teaching Hospitals In Central Ethiopia,2024 (n = 422).**

| Variables | Categories | Prolonged LOS | | COR(95%CI) | AOR(95%CI) | p-value |
|---|---|---|---|---|---|---|
| | | Yes | No | | | |
| Age in years | < 5 | 40 | 122 | 1 | 1 | |
| | 5-10 | 34 | 149 | 0.69(0.41,1.16) | 0.74(0.39,1.38) | 0.350 |
| | >10 | 9 | 68 | 0.40(0.18,0.88) | 0.62(0.25,1.55) | 0.312 |
| Sex | Male | 51 | 221 | 1 | 1 | |
| | Female | 32 | 118 | 0.85(0.51,1.39) | | |
| Residency | Urbane | 33 | 187 | 1 | 1 | |
| | Rural | 50 | 152 | 1.86(1.14,3.03) | 1.41(0.79, 2.51) | 0.245 |
| Type of admission | Elective | 48 | 147 | 1 | 1 | |
| | Emergency | 28 | 180 | 0.47(0.28,0.79) | 0.30(0.14,0.61) | **0.001*** |
| | Urgent | 7 | 12 | 1.78(0.66, 4.79) | 0.85(0.26,2.76) | 0.792 |
| Health insurance | Has Insurance | 32 | 102 | 1 | 0.92(0.48,1.75) | 0.802 |
| | Has No insurance | 51 | 232 | 0.70 (0.45,1.15) | 1 | |
| Comorbidity | Yes | 5 | 7 | 3.04(0.93, 9.83) | 1.71(0.41,7.05) | 0.453 |
| | No | 78 | 332 | 1 | 1 | |
| Preoperative anemia | Yes | 8 | 16 | 2.17 (0.89,5.27) | 1.04(0.34,3.17) | 0.940 |
| | No | 74 | 322 | 1 | 1 | |
| Nutritional status | Normal | 65 | 312 | 1 | 1 | |
| | Undernourished | 18 | 27 | 3.12(1.66, 6.15) | 2.60(1.11, 6.08) | **0.027*** |
| admission diagnosis | GI disorders | 26 | 145 | 1 | 1 | |
| | Anomaly | 22 | 111 | 1.10(0.59,2.05) | 0.52(0.22,1.22) | 0.135 |
| | Trauma | 18 | 40 | 2.50(1.25,5.03) | 4.1(1.75, 9.83) | **0.001*** |
| | Urology | 4 | 10 | 2.23(0.65,7.65) | 1.45(0.33,6.27) | 0.616 |
| | Others | 13 | 33 | 2.19(1.02, 4.72) | 1.33(0.51,3.43) | 0.552 |
| WHO checklist | Present | 50 | 244 | 0.58(0.35,0.97) | 0.69(0.33,1.42) | 0.318 |
| | Absent | 33 | 95 | 1 | 1 | |
| Postop complication | Yes | 29 | 28 | 5.96(3.29,10.8) | 7.8(3.99,15.6) | **0.001*** |
| | No | 54 | 311 | 1 | 1 | |

AOR: adjusted odds ratio; COR: crude odds ratio; LOS: length of stay; GI: Gastrointestinal; WBC: white blood count; WHO; World Health Organization
*significant at a p-value <0.05 and ** significant at a p-value <0.001 level (¥ burn, cellulitis,birantumor,abcess, hernia)

was rapid(Tonsillectomy) and the operational definition utilized (more than 24 hrs) [45].. Nevertheless, the prevalence was lower than a study conducted in China (29.5%) [46]. This might be due to the operational definition used, or might be due to lung surgeries are delicate and require longer follow up days [47–49].

This study revealed that, the median duration of LOS was 6 days which is nearly comparable with the study reported from Tanzania (7 days) [2]. The finding of this study was higher than a study conducted in southern Iran and Saudi Arabia, where, the overall median of LOS in hospitals was reported as 3 and 2.8 days respectively [7,50]. The variation might be due to differences in hospital setup, quality of care, human power, study population, high frequency of health care infection, increased duration of consultation, prolonged waiting time, cause and/or type of surgery, inadequate provision of diagnostic and therapeutic services during health care delivery [51–53].

However, the LOS in this study was found to be lower than in a study conducted among pediatric surgical patients admitted to Arbaminch General Hospital, Southern Ethiopia (23.11 days) [4]. This might be due to differences in the study

setting & surgery specialist volume, availability of diagnostic modalities, admission diagnoses (trauma, burn and fracture was predominant than acute abdomen in this study's setup), and the prevalence of surgical site infection due to differences in quality of care [3,54,55].The finding is also lower than the study reported in China (27.11 days) [56]. This variation could be due to the fact that, high proportion of patients in China were admitted to the pediatric surgical wards as a result of burns [56].This is because, patients admitted due to burn with total body surface area (TBSA) ≥20% are highly prone to infection, which can lead to prolonged hospital stays [53].

The study showed that, within the emergency admission type, the likelihood of PLOS will be decreased compared to the elective admission type. The finding is supported by a study conducted in Tehran, Iran [57]. This might be possibly due to emergeny conditions acute abdomen, mainly appendectomy, which is predominant in our set up, takes lesser hospital stay and elective surgeries, mainly congenital anamalies, which is predominant in our set up, took longer hospital stays. A study done at Arbaminch General Hospital shows emergency admission reduced the length of stay but failed to provide a significant association after controlling for other factors [4]. This might be due to the fact that,children admitted by elective or scheduled admission may have frequent health facility visits with the possibility of having chronic problems,preoperative hospitalstay at the ward for a longer duration, which can increase the risk of hospital-acquired infections(HAIs) [58].

Children with undernourished nutritional status were about three times more likely to stay longer in the hospital than well-nourished children. The finding is inline with a study conducted in theTigray Region, Northern Ethiopia [59]. This might be explained by the fact that, proper nutrition is essential for healing and recovery, and undernourished children may experience delays in wound healing and an increased risk of infection, which can lead to complications during recovery [60,61]. Early nutritional screening and post operative monitoring and management could significantly lower the unintended LOS in the hospital. Hospital administrators could plan for nutrional based management protocols for paediatrics clients that under going surgery [62].

Children admitted due to trauma had four-fold higher odds of a prolonged hospital stay. This finding was also noted in a study at Arbaminch General Hospital [4]. This could be attributed to the complex nature of injuries that can further lead to disability [63]. Moreover, the emotional impact of trauma can also prolong a child's hospital stay [11,64].Rapid assessment, stabilization and early initiation of rehabilitation care could minimize the length of stay. Which necessitates extensive treatment and rehabilitation care [65].

Postoperative complications prolonged the likelihood of hospital stay by eight times higher than those without postoperative complications [62]. This finding was supported by a study in Arsi Zone, Oromia, Ethiopia [11], Arbaminch General Hospital [4],Tanzania [2] and Saudi Arabia [50]. This might be due to, postoperative complications increase the need for additional medical interventions & care demand, leading to longer recovery periods, higher healthcare costs and risk of hospital readmission [54,64,66,67]. Early identification and management of these complications are crucial to mitigate the devastating impact of prolonged hospital stays and poor surgical outcome [68]. Initiatives need to be carried out to prevent and mitigate potential complications and to deliver the intended perioperative care.Even, modest reduction can lead to the improvement of outcomes and cost saving across all health care system in the country both in moderate and high complexity operations [69].

Overall, this study provides an important implication for clinical care, health care management, and further research in the areas of pediatric surgery treatment outcomes such as length of hospital stay.

Healthcare workers could identify prognostic factors associated with prolonged length of stay among pediatrics who have undergone surgical intervention and admitted to pediatric surgical unit for recovery. Healthcare managers could use this evidence to assess and improve the quality of care provided by clinical practitionars. Researchers could be encouraged to conduct further studies on this critical issue to develop effective strategies to predict and reduce prolonged length of hospital stay. The finding can also have further implication for optimal resource utilization,increasing bed turnover rate, reduction of operation cancelation and improving the overall treatment outcomes [48,69].

## Limitations of the study

Due to the nature of the study design(cross-sectional), it does not establish a strong causal association. Additionally, the study does not consider the impact of psychosocial factors, and some other detail treatment protocols due to the reliance on medical records; moreover,differences in measurement approaches used across various studies may lead to an over-estimation or underestimation of the length of hospital stays.

## Conclusion

The study revealed that, approximately one-fifth of the participants experienced prolonged LOS. The median length of hospital stay was 6 days and the mean LOS was 8.6 days. Emergency admission, under-nutrition, admission diagnoses (Trauma-related), and postoperative complications were found to be significantly associated with prolonged LOS. Thus, to achieve the intended treatment outcome early in time,clinicians,hospital administrators,policymakers and other stakeholders should emphasize on children, presented with derangement of their nutritional status, trauma-related surgical problems, and other nonurgent surgical conditions. Furthermore, reducing postoperative complications through expertise quality of care can significantly reduce prolonged hospital lengths of stay among pediatric population admitted in hospital due to surgical conditions. Moreover,ensuring all children to have food security may result in shorter stay in the hospital. Expansion of surgical work force could also improve patient outcome following surgery among paediatrics population.

## Acknowledgments

We thank Haramaya University for its technical support. We would also like to extend our gratitude to the head and staff of AHMC and ARTH, data collectors, and supervisors for their collaboration.

## Author contributions

**Conceptualization:** Redwan Nesha.

**Data curation:** Fentahun Meseret, Redwan Nesha, Haymanot Mezmur, Yohannes Baye, Henok Legesse.

**Formal analysis:** Fentahun Meseret, Redwan Nesha, Haymanot Mezmur, Yohannes Baye, Mulualem Keneni, Ayichew Alemu, Yalew mossie, Henok Legesse.

**Funding acquisition:** Redwan Nesha.

**Investigation:** Fentahun Meseret, Redwan Nesha, Haymanot Mezmur, Yohannes Baye, Henok Legesse.

**Methodology:** Fentahun Meseret, Redwan Nesha, Haymanot Mezmur, Yohannes Baye, Mulualem Keneni, Ayichew Alemu, Yalew mossie, Henok Legesse.

**Resources:** Fentahun Meseret, Redwan Nesha, Haymanot Mezmur, Yohannes Baye, Henok Legesse.

**Software:** Fentahun Meseret, Redwan Nesha, Haymanot Mezmur, Yohannes Baye, Henok Legesse.

**Supervision:** Fentahun Meseret, Haymanot Mezmur, Yohannes Baye, Mulualem Keneni, Ayichew Alemu, Yalew mossie, Henok Legesse.

**Validation:** Fentahun Meseret, Redwan Nesha, Haymanot Mezmur, Yohannes Baye, Mulualem Keneni, Ayichew Alemu, Yalew mossie, Henok Legesse.

**Visualization:** Fentahun Meseret, Redwan Nesha, Haymanot Mezmur, Yohannes Baye, Mulualem Keneni, Ayichew Alemu, Yalew mossie, Henok Legesse.

**Writing – original draft:** Fentahun Meseret, Redwan Nesha, Haymanot Mezmur, Yohannes Baye, Yalew mossie, Henok Legesse.

**Writing – review & editing:** Fentahun Meseret, Redwan Nesha, Haymanot Mezmur, Yohannes Baye, Mulualem Keneni, Ayichew Alemu, Yalew mossie, Henok Legesse.

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
