## [Decision Letter · Decision Letter 0]

31 Jan 2025

Thank you for submitting your manuscript to PLOS ONE. After careful consideration, we feel that it has merit but does not fully meet PLOS ONE’s publication criteria as it currently stands. Therefore, we invite you to submit a revised version of the manuscript that addresses the points raised during the review process.

We look forward to receiving your revised manuscript.

Kind regards,

Abdene Weya Kaso, MPH

Academic Editor

PLOS ONE

Journal Requirements:

http://ir.haramaya.edu.et/hru/bitstream/handle/123456789/6673/Final%20thesis%20after%20Defence.pdf?isAllowed=y&sequence=1

In your revision ensure you cite all your sources (including your own works), and quote or rephrase any duplicated text outside the methods section. Further consideration is dependent on these concerns being addressed.

3. Please note that funding information should not appear in the Acknowledgments section or other areas of your manuscript. We will only publish funding information present in the Funding Statement section of the online submission form. Please remove any funding-related text from the manuscript. 

4. We note that your Data Availability Statement is currently as follows: 

“All relevant data are within the manuscript and its Supporting Information files.”

7. We note you have included a table to which you do not refer in the text of your manuscript. Please ensure that you refer to Tables 2 and 4 in your text; if accepted, production will need this reference to link the reader to the Table.

**Additional Editor Comments:**

The manuscript has been evaluated by two reviewers, and their comments are available below.

The reviewers have raised a number of major concerns. They request language edition and additional information on methodological aspects of the study such as operational definition of the outcome variable and terms.

Reviewers' comments:

Reviewer's Responses to Questions

**Comments to the Author**

1. Is the manuscript technically sound, and do the data support the conclusions?

Reviewer #1: Partly

Reviewer #2: Partly

2. Has the statistical analysis been performed appropriately and rigorously?

Reviewer #1: Yes

Reviewer #2: Yes

3. Have the authors made all data underlying the findings in their manuscript fully available?

Reviewer #1: Yes

Reviewer #2: Yes

4. Is the manuscript presented in an intelligible fashion and written in standard English?

Reviewer #1: No

Reviewer #2: No

Reviewer #1:Comments:

Thank you for the opportunity to review this manuscript, which focuses on the prolonged length of stay among pediatric surgical patients in two hospitals in Ethiopia. I have outlined my feedback as follows:

General Comments:

There are various grammatical, typographical, and coherence issues throughout the document that warrant revision and reorganization of the paragraphs for better clarity and flow.

1. Title:

I suggest refining the title as:

“Prolonged Length of Hospital Stays and Its Associated Factors Among Pediatric Surgical Patients Admitted to Two Teaching Hospitals in Central Ethiopia: A Cross-Sectional Study.”

2. Background:

- The authors state that no previous study has been conducted on this subject in the study setting. However, this argument appears weak as a justification for conducting the study.

- Additionally, various factors influencing the length of hospital stays are mentioned in the background section, and these overlap with the current study's findings. This raises concerns about the originality and potential contribution of the paper to the existing literature.

- I recommend clearly articulating the research gap and providing a strong justification for why this study is necessary.

3. Methodology:

- The authors mention the use of a simple random sampling technique to recruit participants. However, considering the prospective sampling of patients admitted at various times, it is unclear how the sampling frame was established and how computer-generated random sampling was conducted.

Outcome Variable:

- It is unclear why the 75th percentile was chosen as the cutoff point for defining prolonged stays.

- The reference to the 75th percentile under the "Data Processing and Analysis" section needs clarification. Since this metric is based on the length of stay for all participants, its applicability to similar contexts may raise concerns about transferability.

- "No Risk" and "At Risk" Variables:

- The classification of these variables is unclear and requires further explanation, along with appropriate citations.

- Translation of Tools:

- The translation process for the data collection tools is mentioned twice under the "Data Collection Method" and "Data Quality Control" sections. This redundancy should be addressed.

- Data Processing and Analysis:

- Readers may benefit from a clearer explanation of how normality and other assumptions for binary logistic regression were addressed. While multicollinearity is mentioned, further elaboration would enhance clarity.

- There is a duplicate statement about the assessment of the model's goodness of fit using the Hosmer and Lemeshow test, and the multicollinearity evaluation using the Variance Inflation Factor (VIF). Streamlining this information would improve readability.

4. Results:

- This section requires significant revision for grammar and coherence to improve its readability and clarity.

5. Discussion:

- The discussion primarily compares and contrasts the current findings with previous studies, providing justifications for similarities and differences.

- However, it is important to include practical recommendations for practice, policy, and research, particularly in the context of the Ethiopian healthcare system.

Reviewer #2: Comment

I appreciate you, editor, asking me to review the manuscript "Prolonged Length of Hospital Stays and its Associated Factors Among Paediatric Surgical Patients Admitted in Adama Hospital Medical College and Asella Referral and Teaching Hospital, Oromia, Ethiopia: A cross-sectional study based on logistic regression analysis." The study will significantly impact paediatric surgical patients' management, and the length of hospital stay, and related factors are an important metric for assessing hospital care, guaranteeing quality, accessibility, and supplementing resources. Despite this inference, I have provided a few remarks and recommendations that the authors should consider before publication in PLOS ONE.

Abstract

- Please ensure that all acronyms are defined on first mention.

- Better to summarize the background part = make it sound.

- on result part Does emergency admission results in longer hospital stay? - the AOR of 0.30, doesn't signify about that, rather it has protective effect.

- on conclusion - emphasizing children's cause of admission? what does it mean?? or be specific?

Background

- The author concentrated on the unmet need for surgical care in paragraph four of the background section. Does it connect to a hospital stay? If so, it would be better to concentrate on research that shows the percentage of paediatric surgery patients who have a prolonged hospital stay.

- The authors indicated in the final paragraph of the background section that limited study was the reason for conducting the study, but this is insufficient. I suggest the authors address the gaps in the study setting and other justifications for conducting the study.

Methods

- add full stop for the statement of study population.

- Measurement for some variables is required. e.g. comorbidity??, prophylaxis medication??, medication??, Anamia, Type of operation, Types of anaesthesia, postoperative complication, blood transfusion, and complexity of the surgical procedure, SA physical status classification,Admission diagnosis, Novel Paediatric Comorbidity, Type of comorbidity (non-chronic disease and Chronic disease), Preoperative anaemia, Nutritional status at admission, Wating time to seek medical attention in hospital, Preoperative Platelet count, Preoperative WBC count, Intra and post-operative

characteristics (WHO safe surgical checklist, BUPA scoring, Type of surgery, Prophylaxis, Type

of anaesthesia, Duration of anesthesia,Duration of surgery, Postop complication, Type of complication, Blood transfusion, Wound care per day???

- ACSH vs ATRH vs AHMC - please indicate all this abbreviation correctly., it seems three hospitals.

- the list variables on the data collection section and on the independent variables are not in line.

- ASA classification??, Magnitude of the surgery (BUPA scoring)?? how? clearly define and cite it.

-HMIS? BSC?? - define at first instant.

- MUST?? BMI?? how did you assess and classify it.

- I don't think so pre -test was done?? by looking all those variables? not defined well.

- Prolonged length of hospital stay (PLOS). - define at first instant.

- on statistical analysis part the sentence '' Statistical significance was determined at a p-value of less than 0.05 with adjusted odds ratio and 95% of CI' was mentioned three times - please avoid redundancy.

- please revise the statistical analysis part so as to avoid the redundant sentences.

- 100% response rate?? your justification??

Result

- Error! Reference source not found.) on the Preoperative Characteristics?? please also clearly define some variables listed with abbreviation.

- when describing the proportion of sub-categories, please state the magnitude along with percentage. Is not clear about GI disorder, types of comorbidities, and others? for example, malnutrition only 6 out 422 or 12 (comorbidity).

- all the result part needed through revision and correct the table citations.

- The study's reporting and measurement of the length of hospital stay are inconsistent. 75% was the cut-off figure used by the authors to measure the length of hospital stay. However, they reported and discussed with the median. Why? Please modify it.

- Nutrition status?? vs comorbidity (malnutrition)? - multicollinearity

- urology, and anomaly? on factors table? what does it mean - Urology- the branch of medicine that focuses on surgical and medical diseases of the urinary system and the reproductive organs.?? it's not a disease.

- Urology (4) on factor table- < 5, does it violate assumption of chi-square??

* The study's discussion, conclusion, limitations, and strengths are all too shallow and require modification. Furthermore, the recommendation is too shallow and should be revised again.

422? Is it large?

- What about discussion based on the percentage/magnitude??

- What about using variable measurement as a constraint?

- limitation on quality of the data?

- Complete and thorough revision of the manuscript to remove typos and grammatical errors.

**Do you want your identity to be public for this peer review?** For information about this choice, including consent withdrawal, please see our Privacy Policy

Reviewer #1: No

Reviewer #2: No

---

## [Author Response · Author response to Decision Letter 1]

26 Feb 2025

Title: Prolonged Length of Hospital Stays and Its Associated Factors Among Pediatric Surgical Patients Admitted to Two Teaching Hospitals in Central Ethiopia: A Cross-Sectional Study

From: Authors

To: PLOS ONE

Version: 1

Point -by-point response by authors

Sections Independent Review Report Response to comments by Authors

Reviewer #1

General comments There are various grammatical, typographical, and coherence issues throughout the document that warrant revision and reorganization of the paragraphs for better clarity and flow.

Dear respected reviewer, we thank you for your genuine comments. The comments are helpful to increase the quality our study.

Title I suggest refining the title as:

“Prolonged Length of Hospital Stays and Its Associated Factors Among Pediatric Surgical Patients Admitted to Two Teaching Hospitals in Central Ethiopia: A Cross-Sectional Study.”

Thank you for your comments, refining based on your recommendation

Background The authors state that no previous study has been conducted on this subject in the study setting. However, this argument appears weak as a justification for conducting the study.

- Additionally, various factors influencing the length of hospital stays are mentioned in the background section, and these overlap with the current study's findings. This raises concerns about the originality and potential contribution of the paper to the existing literature.

- I recommend clearly articulating the research gap and providing a strong justification for why this study is necessary.

… -Thank you for your comments, we have revised it accordingly.

-The various factors mentioned in the introduction part of the manuscript is based the existing literatures even outside the country just for comparison.

-Thank you, revised!

Methods

The authors mention the use of a simple random sampling technique to recruit participants. However, considering the prospective sampling of patients admitted at various times, it is unclear how the sampling frame was established and how computer-generated random sampling was conducted.

Outcome Variable:

- It is unclear why the 75th percentile was chosen as the cutoff point for defining prolonged stays.

- The reference to the 75th percentile under the "Data Processing and Analysis" section needs clarification. Since this metric is based on the length of stay for all participants, its applicability to similar contexts may raise concerns about transferability.

- "No Risk" and "At Risk" Variables:

- The classification of these variables is unclear and requires further explanation, along with appropriate citations.

- Translation of Tools:

- The translation process for the data collection tools is mentioned twice under the "Data Collection Method" and "Data Quality Control" sections. This redundancy should be addressed.

- Data Processing and Analysis:

- Readers may benefit from a clearer explanation of how normality and other assumptions for binary logistic regression were addressed. While multicollinearity is mentioned, further elaboration would enhance clarity.

- There is a duplicate statement about the assessment of the model's goodness of fit using the Hosmer and Lemeshow test, and the multicollinearity evaluation using the Variance Inflation Factor (VIF). Streamlining this information would improve readability.

-Thank you for your insight, this is due to editorial problem face to face interview did not conducted rather this is a one year study retrieved from February 1, 2023, to March 30 with the data extraction period of one month(April 1- 30, 2024). See the revised manuscript on the abstract section.

-Thank you, revised based on the comment

-Thank you, revised based on the comment

- Thank you ….Distribution of data were checked by shapiro-wilk and Kolmogorov tests(for p-value greater than 0.05,the data were considered as normally distributed) and, revised based on the comment

-Thank you, revised based on the comment

Result This section requires significant revision for grammar and coherence to improve its readability and clarity.

-Thank you, revised based on the comment

Discussion The discussion primarily compares and contrasts the current findings with previous studies, providing justifications for similarities and differences.

- However, it is important to include practical recommendations for practice, policy, and research, particularly in the context of the Ethiopian healthcare system.

Thank you, revised based on the comment

Reviewer #2:

General comments I appreciate you, editor, asking me to review the manuscript "Prolonged Length of Hospital Stays and its Associated Factors Among Paediatric Surgical Patients Admitted in Adama Hospital Medical College and Asella Referral and Teaching Hospital, Oromia, Ethiopia: A cross-sectional study based on logistic regression analysis." The study will significantly impact paediatric surgical patients' management, and the length of hospital stay, and related factors are an important metric for assessing hospital care, guaranteeing quality, accessibility, and supplementing resources. Despite this inference, I have provided a few remarks and recommendations that the authors should consider before publication in PLOS ONE.

Dear respected reviewer, we thank you for your genuine comments. The comments are helpful to increase the quality our study.

Abstract

- Please ensure that all acronyms are defined on first mention.

- Better to summarize the background part = make it sound.

- on result part Does emergency admission results in longer hospital stay? - the AOR of 0.30, doesn't signify about that, rather it has protective effect.

- on conclusion - emphasizing children's cause of admission? what does it mean?? or be specific?

-Thank you! Respected reviewers for your genuine comments, now it has been corrected.

- Thank you, revised based on the comment

-Emergency admission was found to be less likely associated with prolonged stay in the hospital; which means that, those patients presented with non-urgent surgical condition has more likely to stay long time.

Background - The author concentrated on the unmet need for surgical care in paragraph four of the background section. Does it connect to a hospital stay? If so, it would be better to concentrate on research that shows the percentage of paediatric surgery patients who have a prolonged hospital stay.

- The authors indicated in the final paragraph of the background section that limited study was the reason for conducting the study, but this is insufficient. I suggest the authors address the gaps in the study setting and other justifications for conducting the study.

-Thank you, revised….the issue here is if there is unmet need, there is deliance of management; if there is deliance, there is also complication of cases and ultimately increases hospital length of stay even after intervention.

-okay, thank you! Corrected! See the re-vised manuscript.

Method - add full stop for the statement of study population.

- Measurement for some variables is required. e.g. comorbidity??, prophylaxis medication??, medication??, Anamia, Type of operation, Types of anaesthesia, postoperative complication, blood transfusion, and complexity of the surgical procedure, SA physical status classification,Admission diagnosis, Novel Paediatric Comorbidity, Type of comorbidity (non-chronic disease and Chronic disease), Preoperative anaemia, Nutritional status at admission, Wating time to seek medical attention in hospital, Preoperative Platelet count, Preoperative WBC count, Intra and post-operative

characteristics (WHO safe surgical checklist, BUPA scoring, Type of surgery, Prophylaxis, Type

of anaesthesia, Duration of anesthesia,Duration of surgery, Postop complication, Type of complication, Blood transfusion, Wound care per day???

- ACSH vs ATRH vs AHMC - please indicate all this abbreviation correctly., it seems three hospitals.

- the list variables on the data collection section and on the independent variables are not in line.

- ASA classification??, Magnitude of the surgery (BUPA scoring)?? how? clearly define and cite it.

-HMIS? BSC?? - define at first instant.

- MUST?? BMI?? how did you assess and classify it.

- I don't think so pre -test was done?? by looking all those variables? not defined well.

- Prolonged length of hospital stay (PLOS). - define at first instant.

- on statistical analysis part the sentence '' Statistical significance was determined at a p-value of less than 0.05 with adjusted odds ratio and 95% of CI' was mentioned three times - please avoid redundancy.

- please revise the statistical analysis part so as to avoid the redundant sentences.

- 100% response rate?? your justification??

Result

- Error! Reference source not found.) on the Preoperative Characteristics?? please also clearly define some variables listed with abbreviation.

- when describing the proportion of sub-categories, please state the magnitude along with percentage. Is not clear about GI disorder, types of comorbidities, and others? for example, malnutrition only 6 out 422 or 12 (comorbidity).

- Thank you, corrected based on the recommendation and you can see the revised manuscript.

- Thank you, the measurement is based on physician diagnosis(which has been already identified from the patient chart) e.g. for those patients having concomitant disease /comorbidity, It has been labeled as ‘yes’ otherwise ‘NO’ and the same is true for other mentioned variables in the comment.

- Thank you, corrected based on the recommendation

- Thank you, revised based on the recommendation

- Thank you, revised based on the recommendation

- Thank you, revised based on the recommendation

- Thank you, this is because medical charts/patient cards have been replaced by another chart when there is missing of relevant data.

- Thank you, revised based on the recommendation

- Thank you, revised based on the recommendation

discussion - all the result part needed through revision and correct the table citations.

- The study's reporting and measurement of the length of hospital stay are inconsistent. 75% was the cut-off figure used by the authors to measure the length of hospital stay. However, they reported and discussed with the median. Why? Please modify it.

- Nutrition status?? vs comorbidity (malnutrition)? - multicollinearity

- urology, and anomaly? on factors table? what does it mean - Urology- the branch of medicine that focuses on surgical and medical diseases of the urinary system and the reproductive organs.?? it's not a disease.

- Urology (4) on factor table- < 5, does it violate assumption of chi-square??

* The study's discussion, conclusion, limitations, and strengths are all too shallow and require modification. Furthermore, the recommendation is too shallow and should be revised again.

422? Is it large?

- Thank you, revised based on the recommendation

- Thank you, we have used 75th percentile to determine the cut of value for prolonged length of stay and was found to be >=12.5…. Even though, there is no accepted standard cutoff point, some studies preferred to use 75th percentile due to minimizing procedure specific variations in LOS and possible outliers;,in addition we have also checked the length of stay with median and found to be 6, measure of interest in day; we have discussed peripherally by median as median is the most meaningful comparison measure for evaluating LOS data.

-Thank you, we have revised it.

- Thank you, revised

- Thank you, revised based on the recommendation

Discusion - What about discussion based on the percentage/magnitude??

- What about using variable measurement as a constraint?

- limitation on quality of the data?

- Complete and thorough revision of the manuscript to remove typos and grammatical errors.

- Thank you, revised based on the recommendation

- Thank you, revised based on the recommendation

- Thank you, revised

- Thank you, it has been now revised based on your recommendation. generally, the comment was found to be help full to increase the quality of this manuscript; thank you once again!

---

## [Decision Letter · Decision Letter 1]

26 Mar 2025

Dear Dr Meseret,

Thank you for submitting your manuscript to PLOS ONE. After careful consideration, we feel that it has merit but does not fully meet PLOS ONE’s publication criteria as it currently stands. Therefore, we invite you to submit a revised version of the manuscript that addresses the points raised during the review process.

**ACADEMIC EDITOR: **

Manuscript needs comprehensive grammatical and language editionsOperational definition of Prolonged length of stay needs clarity and modificationThe discussion also a few modification based on the pertinent finding

We look forward to receiving your revised manuscript.

Kind regards,

Abdene Weya Kaso, MPH

Academic Editor

PLOS ONE

Journal Requirements:

Additional Editor Comments:

Thank you for revision. The reviewers have critically reviewed your manuscript and raised the following major concerns

1. The manuscript needs comprehensive grammatical and language editions

2. The way outcome variables is operationalized was unclear and needs amendment.

Reviewers' comments:

Reviewer's Responses to Questions

**Comments to the Author**

Reviewer #1: All comments have been addressed

Reviewer #2: (No Response)

2. Is the manuscript technically sound, and do the data support the conclusions?

Reviewer #1: Yes

Reviewer #2: (No Response)

3. Has the statistical analysis been performed appropriately and rigorously?

Reviewer #1: Yes

Reviewer #2: No

4. Have the authors made all data underlying the findings in their manuscript fully available?

Reviewer #1: Yes

Reviewer #2: Yes

5. Is the manuscript presented in an intelligible fashion and written in standard English?

Reviewer #1: No

Reviewer #2: No

Reviewer #1: Dear Authors,

Thank you for addressing the comments. All comments from the previous review were addressed except the typographical, grammatical, and punctuation, which needed to be addressed.

Reviewer #2: I appreciate you inviting me to this manuscript once more, editor. Despite some advancements, the paper still requires development before it can be accepted for publication; for this reason, I have highlighted a few areas that the authors should address below.

General comment: In the response letter, the authors didn't provide clear justification for their response or correction for previous comments and suggestions; at least they need to clearly pinpoint the area of amendment.

Abstract

- Is being limited research on this topic in Ethiopia, particularly in the study setting, the only rationale to conduct the study?

- In the method section, at least incorporate the methods used to assess the outcome variables.

- Your result indicated the overall length of stay was found to be 19.6%. What about the percentage of prolonged length of stay?

- Did the emergency admission result in a prolonged length of hospital stay? How? Or at least separate the variables that are positively or negatively related to prolonged length of hospital stay.

- conclusion - Approximately one-fifth of the participants experienced prolonged hospital stays vs how about overall length of stay - please make it consistence with the result part of the abstract.

- the abstract in the system and on the manuscript are slightly different.

- recommendation still is not feasible base on the finding, please at least recommend for those variables specifically.\

- How did you manage healthcare costs? is it your finding?

Introduction

- Dood improvement- still needs further discussion on the rationale/justification or significance of the study.

- Ethiopian context is not well explored.

Methods

- Why were neonates excluded? any justification? Moreover, on ethical consideration you said'' This study did not expose neonates with sepsisto unnecessary risks''? so why if excluded?

- please provide at least schematic presentation for Sampling procedure? to easily visualize how samples were selected from the two hospitals?

- Data Collection Methods - Replace with Data collection Techniques and Procedures - and add more on the procedures that were undertaken during the data collection processes.

- Dependent and independent variables - please summarize the sentence like ''Prolonged length of hospital stay is the dependent variable. whereas socio-demographic-related factors (age, sex, residency, religion, and medical insurance); pre-operative factors (comorbidity, prophylaxis medication, medication, and anaemia); and intraoperative and postoperative factors (type of operation, anaesthesia, postoperative complication, blood transfusion, and complexity of the surgical procedure) were the independent variables''.

- Nutritional status? how did you measure or classify it? please report in the method section and also include in the independent variable list.

- on outcome variable measurement - 75th percentile for the entire study population were used; it is not clear; better to explain so as to not to confuse the readers.

- the questionnaire was pretested 5% of the sample public hospital (Bekoji hospital). Therefore, what modification was done?

- Does checking normality distribution is appropriate for this data type? may be chi-square distribution? logistic regression assumptions? - please consult biostatisticians.

Result, Discussion and conclusion

- some variables were included in the results part that were not in the independent variables lists -please revise and consider it.

- On factors associated with PLOS among paediatric surgical patients - please check the chi-square assumptions as one of the cells of the variables have below 5 - or modify the variable classification or shorten into two or three levels

- Discussion is to shallow - needs further explanation and justification for the discrepancy in the result with the previous study -e.g. Especially discuss regarding difference in the hospital set-up, availability of medical personnel (speciality)

- recommendation still not feasible, please provide as much as feasible recommendations for hospitals and administrator, policy makers based on the findings.

- Limitations of the Study - regarding outcome measurement? is the outcome measured similar with previous comparable studies? over estimation or under estimation might occur.

- Grammer edition is still required.

THank you.

**Do you want your identity to be public for this peer review?** For information about this choice, including consent withdrawal, please see our Privacy Policy

Reviewer #1: No

Reviewer #2: No

---

## [Author Response · Author response to Decision Letter 2]

20 Jun 2025

Reviewer #1

General comments Thank you for addressing the comments. All comments from the previous review were addressed except the typographical, grammatical, and punctuation, which needed to be addressed.

Dear respected reviewer, we thank you for your genuine comments. The comments are helpful to increase the quality our study. The typographical, grammatical issue was tried to correct through critical review and software detection.

Reviewer #2:

General comments .

Reviewer 2

I appreciate you inviting me to this manuscript once more, editor. Despite some advancements, the paper still requires development before it can be accepted for publication; for this reason, I have highlighted a few areas that the authors should address below.

General comment: In the response letter, the authors didn't provide clear justification for their response or correction for previous comments and suggestions; at least they need to clearly pinpoint the area of amendment. Dear respected reviewer, we thank you for your genuine comments. The comments are helpful to increase the quality our study.

Abstract

- Is being limited research on this topic in Ethiopia, particularly in the study setting, the only rationale to conduct the study?

- In the method section, at least incorporate the methods used to assess the outcome variables.

- Your result indicated the overall length of stay was found to be 19.6%. What about the percentage of prolonged length of stay?

- Did the emergency admission result in a prolonged length of hospital stay? How? Or at least separate the variables that are positively or negatively related to prolonged length of hospital stay.

- Conclusion - Approximately one-fifth of the participants experienced prolonged hospital stays vs how about overall length of stay - please make it consistence with the result part of the abstract.

- the abstract in the system and on the manuscript are slightly different.

- recommendation still is not feasible base on the finding, please at least recommend for those variables specifically.\

- How did you manage healthcare costs? is it your finding?

-Thank you!. The study emphasizes on the evidence gap [limited study], geographical gap and gap in the study population (pediatrics). The gap is incorporated and tracked in the attached manuscript.

- Thank you, revised and tracked based on the comment.

- Thank you, this is typographical error. The percentage is for PLOS. It is corrected and tracked.

-Thank you. The emergency admission was negatively related to PLOS. The description [Factors associated] was corrected and tracked according to your suggestion to separate narration.

-Thank you, corrected. We tried to be consistent with abstract summary [conclusion] and the result. The overall LOS was mistakenly included. Only PLOS was tried to be narrated. Its corrected and tracked.

-Thank you. Adjustment was made in the online system.

-Thank you, corrected!. Specific recommendation was provided.

- Thank you. It was an error made in preparing the manuscript. It is removed and tracked.

Introduction .

- Dood improvement- still needs further discussion on the rationale/justification or significance of the study.

- Ethiopian context is not well explored.

-Thank you. Some improvement was made in the rationale and some evidences were incorporated in Ethiopian context.

Method

- Why were neonates excluded? any justification? Moreover, on ethical consideration you said'' This study did not expose neonates with sepsisto unnecessary risks''? so why if excluded?

- please provide at least schematic presentation for Sampling procedure? to easily visualize how samples were selected from the two hospitals?

- Data Collection Methods - Replace with Data collection Techniques and Procedures - and add more on the procedures that were undertaken during the data collection processes.

- Dependent and independent variables - please summarize the sentence like ''Prolonged length of hospital stay is the dependent variable. whereas socio-demographic-related factors (age, sex, residency, religion, and medical insurance); pre-operative factors (comorbidity, prophylaxis medication, medication, and anaemia); and intraoperative and postoperative factors (type of operation, anaesthesia, postoperative complication, blood transfusion, and complexity of the surgical procedure) were the independent variables''.

- Nutritional status? how did you measure or classify it? please report in the method section and also include in the independent variable list.

- on outcome variable measurement - 75th percentile for the entire study population were used; it is not clear; better to explain so as to not to confuse the readers.

- the questionnaire was pretested 5% of the sample public hospital (Bekoji hospital). Therefore, what modification wasdone?

- Does checking normality distribution is appropriate for this data type? may be chi-square distribution? logistic regression assumptions? - please consult biostatisticians.

- Thank you,this is editorial issue, it has been revised; see the revised manuscript.

-Thank you, revised based on your comment. See figure 1on the manuscrpit

-- Thank you, revised based on your comment.

-- Thank you, corrected based on the recommendation

- Thank you, corrected based on the recommendation. See the revised manuscript.

-Thank you. Little explanation is provided and tracked in the attached manuscript.

-Thank you, based on the pretest, some modification was made on the check list…comorbid variables including nutritional status added due to its accessibility on patient medical chart; whereas variables like parental education, occupation, socioeconomic issue and other psychological factors were removed as it was indicated under limitation part of this manuscript.

-thank you for your insight, it is not as such important based on my consultation and removed accordingly.

Result, discussion&coculusion

- some variables were included in the results part that were not in the independent variables lists -please revise and consider it.

- On factors associated with PLOS among paediatric surgical patients - please check the chi-square assumptions as one of the cells of the variables have below 5 - or modify the variable classification or shorten into two or three levels

- Discussion is to shallow - needs further explanation and justification for the discrepancy in the result with the previous study -e.g. Especially discuss regarding difference in the hospital set-up, availability of medical personnel (speciality)

- recommendation still not feasible, please provide as much as feasible recommendations for hospitals and administrator, policy makers based on the findings.

- Limitations of the Study - regarding outcome measurement? is the outcome measured similar with previous comparable studies? over estimation or under estimation might occur.

- Grammer edition is still required.

THank you

- Thank you, revised based on the recommendation

- Thank you. We have tried to prevent CI inflation trough cross-tabulation of each cell. To the best of our knowledge, if the rows of a variable are more than 2, one cell with observed value of less than 5 is acceptable.

-Thank you, we have revised it.

- Thank you, it has been now revised based on your recommendation.

Generally, the comment was found to be help full to increase the quality of this manuscript;

-thank you once again for your genuine comments through out!

---

## [Decision Letter · Decision Letter 2]

15 Jul 2025

Dear Dr. Meseret,

Thank you for submitting your manuscript to PLOS ONE. After careful consideration, we feel that it has merit but does not fully meet PLOS ONE’s publication criteria as it currently stands. Therefore, we invite you to submit a revised version of the manuscript that addresses the points raised during the review process.

We look forward to receiving your revised manuscript.

Kind regards,

Abdene Weya Kaso, MPH

Academic Editor

PLOS ONE

Journal Requirements:

Reviewers' comments:

Reviewer's Responses to Questions

**Comments to the Author**

Reviewer #1: All comments have been addressed

Reviewer #2: (No Response)

2. Is the manuscript technically sound, and do the data support the conclusions?

Reviewer #1: Yes

Reviewer #2: Partly

3. Has the statistical analysis been performed appropriately and rigorously?

Reviewer #1: Yes

Reviewer #2: Yes

4. Have the authors made all data underlying the findings in their manuscript fully available?

Reviewer #1: No

Reviewer #2: Yes

5. Is the manuscript presented in an intelligible fashion and written in standard English?

Reviewer #1: No

Reviewer #2: Yes

Reviewer #1: Review report

Dear authors,

Thank you for the opportunity to review your work once again. The manuscript has shown significant improvement compared to previous drafts. Please find my feedback below:

• Title: the title of the manuscript is clear and descriptive including the settings and design. However, for concise presentation I would recommend modifying as follows” Prolonged length of stays and its associated factors among paediatric surgical patients in central Ethiopia: a cross-sectional study”.

• Abstract:

o Background: in this section the detailed description of this study significance and objectives are mentioned. However, the first three lines seems reptation of already existing evidence. The consecutive lines also worth editing as a reptation of ideas. For instance, this statement can be modified as “However, there is limited evidence in Ethiopia, particularly among paediatrics population”.

Introduction:

• Detailed description of previous literature, significance and rationale of the study described. However, typographical and grammatical editing may enhance the clarity for the readers. In addition, concise presentation is needed for focused presentation.

Methods

• Under inclusion and exclusion criteria authors mentioned that the “ neonates” were excluded. Does this mean that those neonates who underwent surgery excluded? Explicitly mentioning the rationale for exclusion of neonates may enhance the clarity for the readers.

• Sampling: authors mentioned that the simple random sampling was used in the text. However, in the figure 1 “consecutive sampling” was outlined?

• Two teaching hospitals were purposively selected from central Ethiopia. however, it was unclear why two hospitals were selected? Also, it would enhance the clarity if the total number of hospitals/teaching hospitals found in the study settings were mentioned to understand the context.

• Data collection techniques and procedures: Does the incomplete medical records excluded or replaced by the additional sample?

• Data processing and analysis: The author classified the length of hospital stay based on the 75th percentile, However, there was no justification why 75th percentile used or references from previous literature or guidelines?

Reviewer #2: Dear Editor,

I appreciate you asking me to review this work once more. The manuscript has been greatly improved by the authors. However, I have included some of my thoughts and recommendations for the authors to reconsider in order to further enhance the manuscript and prepare it for a worldwide audience:

- On affiliation for the first author—please modify Hawas to Hawassa.

- Better to modify the topic as " Prolonged Length of Hospital Stays and Its Associated Factors Among Pediatric

Surgical Patients Admitted to Teaching Hospitals in Central Ethiopia: A Cross-Sectional Study

- on an abstract part of the conclussion i recommend the authors to provide feasible recommendation that aimd to improve the service given at the hospitals.

- Do the hospitals have equal lengths of time? Is there no difference among them? Are similar interventions recommended to each hospital? What about administration, infrastructure, availability of staff, and medical facility differences among each hospital? This should be clearly identified to have feasible interventions to solve the problem.

- Keywords: Pediatrics, Length of Hospital stay, Factors, , Surgeryt, , Central Ethiopia (Please correct the commas and the words.)

- The introduction section was greatly improved. I recommend at the end of the introduction section to add the significance of the study.

- On methods, add a full stop to the sentence about the study population.

- The authors need to explicitly state the types of data collected. If the data consists entirely of secondary sources or is supplemented by interviews with caregivers or mothers, this should be clearly indicated in the methods section. Additionally, for the secondary data, it is important to explain why a structured questionnaire was used instead of checklists.

- I check the authors response on the use of 5% pretest, and a modification was made based on that, but on the main document, no modification was made, so I encourage the authors to include those modifications made due to the pretest on the main document.

- Rearrange the position of data quality control after the statistical analysis part.

- Is there any justification to report both the median and mean of length of stay? Did you check symmetry?

- Improved results and discussion sections. I recommend that the authors keep consistency and focus on the objectives (prevalence and associated variables)

- better to add a section—the practical implication of the study (on discussion or after it)

- In the Limitation section, consider addressing the differences in measurement approaches used across various studies. Some studies referenced the mean or median, while your study utilized the 75th percentile. Does this create a significant gap when comparing the results? It would be beneficial to discuss this issue more thoroughly in the Discussion section.

- Please expand your recommendation about possible intervention that would be made to address the problems.

- I recommend that the author conduct a comprehensive review of the entire document. There are several inconsistencies in the use of commas, periods, and grammar throughout. Additionally, it would be beneficial to address recommendations that account for local differences among the selected hospitals, such as implementing subgroup analysis.

Thank you.

**Do you want your identity to be public for this peer review?** For information about this choice, including consent withdrawal, please see our Privacy Policy

Reviewer #1: No

Reviewer #2: **Yes: ** Habtamu Endashaw Hareru

---

## [Author Response · Author response to Decision Letter 3]

23 Jul 2025

Reviewer #1

Dear authors,

Thank you for the opportunity to review your work once again. The manuscript has shown significant improvement compared to previous drafts. Please find my feedback below:

AN: Dear respected reviewer, we thank you for your genuine comments. The comments are helpful to increase the quality our study.

• The title of the manuscript is clear and descriptive including the settings and design. However, for concise presentation I would recommend modifying as follows” Prolonged length of stays and its associated factors among pediatric surgical patients in central Ethiopia: a cross-sectional study”.

AN: Thank you, for your comments, we have revised based on your comments. See the revised title on the revised manuscript.

o Background: in this section the detailed description of this study significance and objectives are mentioned. However, the first three lines seems reptation of already existing evidence. The consecutive lines also worth editing as a reptation of ideas. For instance, this statement can be modified as “However, there is limited evidence in Ethiopia, particularly among paediatrics population”.

AN:Thank you, for your comments, we have revised based on your comments. See the revised manuscript.

• Detailed description of previous literature, significance and rationale of the study described. However, typographical and grammatical editing may enhance the clarity for the readers. In addition, concise presentation is needed for focused presentation.

AN:Thank you, for your comments, we have revised the typographical and other grammatical issues. see the revised manuscript.

• Under inclusion and exclusion criteria authors mentioned that the “ neonates” were excluded. Does this mean that those neonates who underwent surgery excluded? Explicitly mentioning the rationale for exclusion of neonates may enhance the clarity for the readers.

• AN :Neonatal and pediatric surgery differ significantly due to the unique physiological characteristics and vulnerabilities of newborns(neonates) compared to older infants and children

• Sampling: authors mentioned that the simple random sampling was used in the text. However, in the figure 1 “consecutive sampling” was outlined?

AN:• Thank you, for your comments, we have revised it now. see the revised figure.

• Two teaching hospitals were purposively selected from central Ethiopia. However, it was unclear why two hospitals were selected? Also, it would enhance the clarity if the total number of hospitals/teaching hospitals found in the study settings were mentioned to understand the context.

AN:• Thank you, for your comments, the two teaching hospitals have been selected purposely than other non-teaching/non referral hospitals found in Oromia region centeral part of Ethiopia;since they are the only hospitals that can give pediatric surgery services in a little more detail in comparison with other low level hospitals located near the catchment area.

• Data collection techniques and procedures: Does the incomplete medical records excluded or replaced by the additional sample?

AN: Incomplete medical records were replaced with additional random sample….and it is indicated under the sub section of data collection techniques and procedures of the manuscript.

• Data processing and analysis: The author classified the length of hospital stay based on the 75th percentile, however, there was no justification why 75th percentile used or references from previous literature or guidelines?

AN: Neonatal and pediatric surgery differ significantly due to the unique physiological characteristics and vulnerabilities of newborns(neonates) compared to older infants and children

• Incomplete medical records were replaced with additional random sample….and it is indicated under the sub section of data collection techniques and procedures of the manuscript.

AN: Thank you, for your comments…all necessary justification and citation have been highlighted under operational definition subsection of the manuscript.

Reviewer #2:

Dear Editor,

I appreciate you asking me to review this work once more. The manuscript has been greatly improved by the authors. However, I have included some of my thoughts and recommendations for the authors to reconsider in order to further enhance the manuscript and prepare it for a worldwide audience:

AN: Dear respected reviewer, we thank you for your genuine comments. The comments are helpful to increase the quality our study.

- On affiliation for the first author—please modify Hawas to Hawassa.

AN: Thank you, we have revised it.

- Better to modify the topic as " Prolonged Length of Hospital Stays and Its Associated Factors Among Pediatric

Surgical Patients Admitted to Teaching Hospitals in Central Ethiopia: A Cross-Sectional Study

AN:Thank you, we have revised it.

- on an abstract part of the conclussion i recommend the authors to provide feasible recommendation that aimd to improve the service given at the hospitals.

- Do the hospitals have equal lengths of time? Is there no difference among them? Are similar interventions recommended to each hospital? What about administration, infrastructure, availability of staff, and medical facility differences among each hospital? This should be clearly identified to have feasible interventions to solve the problem.

AN: -Thank you, we have revised it.

-The finding indicates nearly comparable length of stay with similar prognostic factors; since both are teaching hospitals serving similar population in terms of characteristics and have nearly comparable capacity in service delivery as well.

- Keywords: Pediatrics, Length of Hospital stay, Factors, , Surgeryt, , Central Ethiopia (Please correct the commas and the words.)

AN:Thank you, we have corrected.

- The introduction section was greatly improved. I recommend at the end of the introduction section to add the significance of the study.

AN:Thank you, for your comments, we have revised based on your comments. See the revised manuscript.

- On methods, add a full stop to the sentence about the study population.

AN:- Thank you, we have added now

- The authors need to explicitly state the types of data collected. If the data consists entirely of secondary sources or is supplemented by interviews with caregivers or mothers, this should be clearly indicated in the methods section. Additionally, for the secondary data, it is important to explain why a structured questionnaire was used instead of checklists.

AN:Thank you, for your comments, we have revised based on your comments. See the revised manuscript.

- I check the authors response on the use of 5% pretest, and a modification was made based on that, but on the main document, no modification was made, so I encourage the authors to include those modifications made due to the pretest on the main document.

AN:Thank you, for your comments, we have revised based on your comments. See the revised manuscript.

- Rearrange the position of data quality control after the statistical analysis part.

AN:- Thank you, for your comments, we have revised based on your comments. See the revised manuscript.

- Is there any justification to report both the median and mean of length of stay? Did you check symmetry?

AN:- -Median is more appropriate due to skewness as a result of time variability

Improved results and discussion sections. I recommend that the authors keep consistency and focus on the objectives (prevalence and associated variables)

- better to add a section—the practical implication of the study (on discussion or after it)

AN:Thank you, for your comments, we have revised based on your comments

In the Limitation section, consider addressing the differences in measurement approaches used across various studies.

Some studies referenced the mean or median, while your study utilized the 75th percentile. Does this create a significant gap when comparing the results? It would be beneficial to discuss this issue more thoroughly in the Discussion section

AN:Thank you, for your comments, we have revised based on your comments

Please expand your recommendation about possible intervention that would be made to address the problems

AN:Thank you, for your comments, we have revised based on your comments

- I recommend that the author conduct a comprehensive review of the entire document. There are several inconsistencies in the use of commas, periods, and grammar throughout. Additionally, it would be beneficial to address recommendations that account for local differences among the selected hospitals, such as implementing subgroup analysis.

Thank you.

AN:Thank you, for your comments, we have revised the inconsistencies in use of comma and other typographical issues; additionally, the finding indicates nearly comparable length of stay with similar prognostic factors; since both are teaching hospitals serving similar population in terms of characteristics and have nearly comparable capacity in service delivery as well.

---

## [Editor Report · Decision Letter 3]

25 Jul 2025

Prolonged Length of Hospital Stays and Its Associated Factors Among Pediatric Surgical Patients Admitted to Two Teaching Hospitals in Central Ethiopia: A Cross-Sectional Study

PONE-D-24-56882R3

Dear Mr Fentahun Meseret,

We’re pleased to inform you that your manuscript has been judged scientifically suitable for publication and will be formally accepted for publication once it meets all outstanding technical requirements.

Kind regards,

Abdene Weya Kaso, MPH

Academic Editor

PLOS ONE
---

## [Editor Report · Acceptance letter]

PONE-D-24-56882R3

PLOS ONE

Dear Dr. Meseret,

I'm pleased to inform you that your manuscript has been deemed suitable for publication in PLOS ONE. Congratulations! Your manuscript is now being handed over to our production team.

Kind regards,

on behalf of

Mr. Abdene Weya Kaso

Academic Editor

PLOS ONE